# Vitamin D prescribing in children in UK primary care practices: a population-based cohort study

Mandy Wan,[1,2] Laura J Horsfall,[3] Emre Basatemur,[4] Jignesh Prakash Patel,[2,5] Rukshana Shroff,[6] Greta Rait[3]

RS and GR are joint senior authors.

For numbered affiliations see end of article.

**Correspondence to**
Mandy Wan;
Mandy.Wan@gstt.nhs.uk

## ABSTRACT

**Objective** To examine temporal changes in the incidence and patterns of vitamin D supplementation prescribing by general practitioners (GPs) between 2008 and 2016.

**Design** Population-based cohort study.

**Setting** UK general practice health records from The Health Improvement Network.

**Participants** Children aged 0 to 17 years who were registered with their general practices for at least 3 months.

**Outcome measures** Annual incidence rates of vitamin D prescriptions were calculated, and rate ratios were estimated using multivariable Poisson regression to explore differences by sociodemographic factors. Data on the type of supplementation, dose, dosing schedule, linked 25-hydroxyvitamin D (25(OH)D) laboratory test results and clinical symptoms suggestive of vitamin D deficiency were analysed.

**Results** Among 2 million children, the crude annual incidence of vitamin D prescribing increased by 26-fold between 2008 and 2016 rising from 10.8 (95% CI: 8.9 to 13.1) to 276.8 (95% CI: 264.3 to 289.9) per 100 000 person-years. Older children, non-white ethnicity and general practices in England (compared with Wales/Scotland/Northern Ireland) were independently associated with higher rates of prescribing. Analyses of incident prescriptions showed inconsistent supplementation regimens with an absence of pre-supplementation 25(OH)D concentrations in 28.7% to 56.4% of prescriptions annually. There was an increasing trend in prescribing at pharmacological doses irrespective of 25(OH)D concentrations, deviating in part from UK recommendations. Prescribing at pharmacological doses for children with deficient status increased from 3.8% to 79.4%, but the rise was also observed in children for whom guidelines recommended prevention doses (0% to 53%). Vitamin D supplementation at pharmacological doses was also prescribed in at least 40% of children with no pre-supplementation 25(OH)D concentrations annually.

**Conclusions** There has been a marked and sustained increase in vitamin D supplementation prescribing in children in UK primary care. Our data suggests that national guidelines on vitamin D supplementation for children are not consistently followed by GPs.

## Strengths and limitations of this study

► This is the first population-based study to examine the extent and variability of vitamin D supplementation prescribing practice using representative primary care data.

► This study does not contain data from secondary care, however, vitamin D deficiency is largely managed in the primary care setting.

► Although several explanations for the factors that might have contributed to changes in vitamin D supplementation prescribing over time are provided, it was not possible to determine the practitioners' rationale for prescribing, nor could we assess the clinical appropriateness of prescriptions on an individual level.

## INTRODUCTION

Vitamin D is an essential prohormone that humans obtain through cutaneous photosynthesis, diet or supplementation. While few would argue the importance of its physiological role in maintaining calcium homeostasis and bone mineralisation, the beneficial effects of vitamin D supplementation is a topic of much controversy.[1–5] In children, severe vitamin D deficiency can cause rickets and hypocalcaemic seizures, but the clinical consequences of vitamin D insufficiency are less established despite the expanding literature in this area. Findings from numerous epidemiological studies have linked low 25-hydroxyvitamin D (25(OH)D) concentrations to an increased risk of a myriad of adverse health consequences in both adults and children,[6 7] where low concentrations in children have been linked to asthma,[8 9] eczema,[10] respiratory tract infections[11 12] and diabetes,[13] among others.[6] Yet, the majority of randomised controlled trials do not show improved outcomes with vitamin D supplementation,[6 7 14 15] and meta-analyses are inconclusive and widely debated.[6–8 11 12 14–18]

Against this backdrop of ongoing debate, vitamin D deficiency is widely prevalent in all age groups and seen as a growing public health concern worldwide.[19] The US,[20] Canada,[21] Australia,[22 23] France[24] and the UK[25 26] have all reported a significant increase in laboratory vitamin D testing, with as much as a 90-fold increase over a 10-year period.[22 23] In England, there has also been a marked increase in the number of vitamin D prescriptions issued by general practitioners (GPs) in primary care, with annual spending on vitamin D prescriptions increasing from £0.5 to £40 million between 2007 and 2016,[27] a rising trend that was partly attributed to vitamin D prescriptions issued for children.[28]

These reported trends have understandably led many to question the clinical appropriateness of vitamin D testing in practice.[29 30] However, even though vitamin D is also widely available without a prescription, the increasing number of GPs issuing vitamin D prescriptions has surprisingly not attracted the same degree of attention. For the UK where the healthcare service is publicly-funded and free, the national strategy is centred on primary prevention in managing vitamin D deficiency. Specifically, UK national guidance (a summary of UK guidelines is presented in online supplementary table S1) has, until relatively recently, targeted recommendations on supplementation mainly towards high-risk groups.[31–40] All children under 5 years of age including breastfed infants, if their mother is also at risk of vitamin D deficiency, were recommended to take daily vitamin D supplement at prevention doses.[32–35 37 38] There were no specific recommendations for children 5 years or above unless individuals were considered at risk. In any children where 25(OH)D is tested, treatment at pharmacological doses is guided by whether 25(OH)D concentration is below 25 nmol/L, the threshold set by the UK Scientific Advisory Committee on Nutrition as vitamin D-deficient.[31 39] Little is known about how well these recommendations are implemented in practice, but it has been noted that the complexity of the advice and poor communications may have caused confusion among both health professionals and the public.[37] Considering there are numerous other international and national guidelines on vitamin D supplementation and health which differ in their definition of vitamin D deficiency and supplementation recommendations,[41–43] a comprehensive analysis of vitamin D prescribing by GPs offers an opportunity to facilitate the appropriate use of vitamin D supplementation for patient benefit and contain healthcare expenditure.

The aim of our study was to examine recent trends in prescribing patterns of vitamin D supplementation in children. We quantified temporal changes in the incidence of vitamin D prescriptions issued by GPs in a UK population-based study. Moreover, we examined the proportions of prescriptions by type of supplementation, dose and dosing schedule, as well as the proportions that can be linked to 25(OH)D laboratory test results and clinical symptoms suggestive of vitamin D deficiency.

## METHODS

### Data source

We obtained electronic health records from patients registered with UK general practices contributing to The Health Improvement Network (THIN). The THIN database contains anonymised data from 744 general practices using the Vision computer system (In Practice Systems, London, UK). Data are from all patients in participating practices unless individual patients opt out of THIN. It contains data of approximately 16 million patients which has been shown to be broadly representative of the UK population in terms of age, sex, prevalence of medical conditions and mortality rates.[44] As of 2015, the THIN dataset is reported to cover 6% of the UK population.[45]

THIN contains information on diagnoses, laboratory tests, symptoms and hospital referrals, as well as basic sociodemographic information. Clinical events and diagnoses are recorded using a hierarchical coding system called Read codes.[46] THIN also includes records of all prescriptions issued and these are linked to the British National Formulary. Prescribing data are particularly well recorded in THIN since the computerised entry made by the GP is directly used to issue a prescription to the patient. Prescribing data in THIN has been shown to be comparable to data on dispensed prescriptions with a mean practice redemption rate for all prescribing of 98.5%.[47]

### Study population

Study cohort selection is presented in online supplementary figure S1. We identified a cohort of children aged 0 to 17 years registered with a THIN general practice from 1 January 2008 to 31 December 2016. Individuals were not eligible to enter the study until 3 months after their registration with the practice in order to exclude those with prevalent prescriptions.[48] Cohort entry was defined by the latest of: the study start date (1 January 2008); 3 months after the patient's registration with the practice; or the date the practice met two predefined quality indicators (acceptable mortality recording and acceptable computer usage) for electronic data recording.[49 50] Children with cystic fibrosis, chronic renal failure, inflammatory bowel diseases or chronic liver diseases were excluded as these conditions are known to affect vitamin D absorption or metabolism and therefore require specialist management (code lists for these conditions were created using a previously published methodology).[51] Children were followed until an incident prescription of vitamin D, or censored on death, midpoint of their 18th birthday, end of registration with a practice, the last collection date of data from the practice or end of the study period (31 December 2016), whichever occurred first. The total number of patient-years between cohort entry and exit defined the denominator for incidence calculations.

### Outcome

A vitamin D prescription was defined as a first prescription record (incident prescription) of either: a

single-ingredient product containing calciferol, ergocalciferol or colecalciferol as the only active ingredient; or a combination product containing calciferol, ergocalciferol or colecalciferol with other active ingredients (eg, ergocalciferol with calcium, multi-vitamin preparations). For the latter, only those that could be linked to a Read code indicating vitamin D deficiency or insufficiency or 25(OH)D testing within 90 days either side of the date of the prescription were included. Comprehensive lists of drug codes and Read codes were developed using a previously published methodology (online supplementary tables S2 and S3).[51] Incident cases were defined as children with a first ever recording of a vitamin D prescription in THIN during the study period. The date at which the first recording of vitamin D prescription was made was classified as the index date. Consultations where GPs did not issue a prescription but recommended self-purchase of over the counter vitamin D supplement were not considered as prescriptions.

## Covariates

Age groups were defined as: up to 6 months; 6 months – 4 years; 5–11; and 12–17 years. Ethnicity was grouped into the 2011 UK Census 5-category classification: children with multiple ethnicity records belonging to different categories were included in the analysis under the 'missing data' category. Townsend index, an area-based relative measure of material deprivation, was extracted: the value closest to the start of the observation period was used for patients with multiple records. For this exploratory study, a pragmatic approach was taken to handle missing data separately under the 'missing data' category.

25(OH)D laboratory test results associated with incident prescriptions were identified by temporal proximity. A result was considered to be contemporary and would have formed the basis of prescribing if it occurred 90 days prior to the index date. Based on the elimination half-life of 25(OH)D, results recorded between 91–180 days after the index date were used to assess post-supplementation vitamin D status. Concentrations<25 nmol/L were categorised as deficient as per the UK Department of Health and Social Care definition relevant at the time of the study period.[31] Concentrations were further categorised as insufficient (25–50 nmol/L) as defined by the British Paediatric and Adolescent Bone Group and supported by the British Society of Paediatric Radiology and the Royal College of Paediatrics and Child Health.[34 36 38]

Read codes for symptoms or clinical complications that could potentially be related to vitamin D deficiency within the 90 days prior to the index date were extracted and organised into 12 categories as presented in online supplementary table S4.

## Incident vitamin D prescriptions analyses

Children with multiple vitamin D prescriptions on the index date (534 patients) were excluded from prescription pattern analyses as it was not possible to determine whether all prescriptions were relevant. Prescriptions were categorised according to the type of supplementation as per case definition above. Equivalent daily dose (EDD) was calculated for each incident prescription as determined by the quantity of vitamin D (international unit (IU)) prescribed per dose and the prescribed dosing frequency. Prescriptions with titration dosages (eg, two tablets daily for 2 weeks and then one tablet monthly) were handled by considering the first dosage only to capture the initial management strategy. Where unit of dosing was missing (eg, every day), one single dosing unit was assumed for tablet/capsule products only. Generic (eg, as directed) or ambiguous (eg, once daily for a liquid product) dosages were categorised under the 'undetermined' category. EDDs were further categorised into dose bands. To assist interpretation, doses of 280–400 IU were considered prevention doses across all age groups based on recommendations relevant at the time of the study period.[31 33–35 39] Doses between 401–1000 IU were categorised as a separate prevention dose category given divergent views exist within the medical community with regards to supplementation strategies for primary prevention.[36–38] Doses between 1001 and 10 000 IU/day were considered as pharmacological treatment doses independent of age.[36 38] Prescription quantity supplied in days was calculated based on total prescription quantity issued and corresponding dosage and capped at 365 days to take account of product shelf life. High-dose vitamin D regimens (stoss regimens) prescribed either as intramuscular injections or oral preparations were categorised separately and reported as either low dose (total dose <1 50 000 IU) or high dose (total dose ≥1 50 000 IU) stoss. As oral stoss regimens may be prescribed as a single dose or spread over several days, prescriptions with explicit duration directions for ≤15 days (eg, 5000 IU once daily for 14 days), or where the quantity prescribed were ≤7 days with an equivalent daily dose ≥10 000 IU (eg, 5 capsules of 20 000 IU with dosing direction 'one capsules once daily') were also categorised as stoss.

## Statistical analysis

Cohort characteristics are presented by sex, ethnicity, socioeconomic quintile, country, calendar year and year of cohort entry as frequencies (%). Medians and IQR are presented for age at entry and follow-up time. We calculated crude annual incidence rates by dividing the number of children with a first ever recording of a vitamin D prescription by the Person Years at Risk. Confidence intervals (95% CI) were calculated assuming a Poisson distribution. Using Poisson regression, we estimated sex-stratified incidence rate ratios with 95% CI, adjusting for potential confounders. The time period was fitted as both a linear and categorical variable and models compared using the log likelihood ratio test: there was evidence that the categorised time period was a better fit with the data and thus the time period was treated as a categorical variable. Age was also fitted as a categorial variable as it had a non-linear relationship. A fully adjusted sex-stratified model with an interaction between age and ethnicity

was fitted and examined: while the interaction term was statistically significant, it did not change the estimates of the other variables and the incidence rates predicted by the two models were similar. However, as the model with the interaction term resulted in unstable predictions, the interaction term was not included in the final model. The final model with the inclusion of the GP practice as a random effect to account for any data clustering was used to calculate adjusted incidence rate ratios (IRRs). Proportions of prescriptions by type of supplementation, dose and dosing schedule, linked to 25(OH)D laboratory test concentrations and clinical symptoms suggestive of vitamin D deficiency, are presented as frequencies (%). Trends in prescribing characteristics were quantified using Spearman correlation coefficients using all years between 2008 and 2016. Data analysis was conducted using Stata 15 (Stata Corp, College Station, Texas, USA).

Three sensitivity analyses were performed. First, we considered a narrower case definition for vitamin D prescription, where only single-ingredient products were included. Second, we repeated the incidence analysis using complete cases only, that is among children who had complete data on sex, age, ethnicity, socioeconomic status, country and calendar year. Finally, we considered a broader definition for prevention doses, where equivalent daily doses of ≤1000 IU were included.

### Patient and public involvement

Patient and parent/carer representatives were involved in developing study plans and approving the lay summaries for the grant supporting this study. Patients were not involved in setting the specific research question, the outcome measure, the design or implementation of this study. No patients were asked to advise on interpretation or writing up of results. It is not possible to disseminate the results of the research to study participants, but views from patient and parent/carer representatives will be sought in disseminating the research findings.

### Ethics

The THIN data collection scheme to obtain and provide anonymised patient data was approved by the National Health Service South-East Multicentre Research Ethics Committee (REC Reference: 04/MRE01/9) in 2002. The study protocol was approved by an independent Scientific Review Committee (Reference: 16THIN035).

### RESULTS

A total of 2 051 403 eligible children from 723 general practices contributed data between 1 January 2008 and 31 December 2016, giving a total of 8 million person-years of follow-up. Of those children, 12 277 had an incident vitamin D prescription during the study period. Patient characteristics stratified by sex, ethnicity, country of general practice and socioeconomic status are shown in table 1 for the overall cohort and the incident cohort.

**Table 1** Descriptive characteristics of the overall study cohort and incident cohort

| | Overall cohort (n=2 051 403) | Incident cohort (n=12 277) |
|---|---|---|
| Median age at entry, years (IQR) | 5.5 (0.8–11.5) | 7.6 (3.5–10.5) |
| Follow-up time, years (IQR) | 3.5 (1.5–6.3) | 4.2 (2–6.1) |
| Sex, n (%) | | |
| Male | 1 055 395 (51.5) | 4936 (40.2) |
| Female | 996 008 (48.6) | 7341 (59.8) |
| Ethnicity, n (%) | | |
| White | 878 360 (42.8) | 2836 (23.1) |
| Asian/Asian British | 76 840 (3.8) | 2928 (23.9) |
| Black/African/Caribbean/ Black British | 47 206 (2.3) | 1275 (10.4) |
| Other ethnic group | 17 656 (0.9) | 312 (2.5) |
| Mixed/multiple ethnic group | 28 605 (1.4) | 295 (2.4) |
| Missing | 1 002 736 (48.9) | 4631 (37.7) |
| Country of GP practice, n (%) | | |
| England | 1 451 923 (70.8) | 10 921 (88.9) |
| Northern Ireland | 84 171 (4.1) | 177 (1.4) |
| Scotland | 310 142 (15.1) | 597 (4.9) |
| Wales | 205 167 (10) | 582 (4.7) |
| Townsend deprivation index quintile, n (%) | | |
| 1 (least deprived) | 397 057 (19.4) | 1452 (11.8) |
| 2 | 357 567 (17.4) | 1377 (11.2) |
| 3 | 396 234 (19.3) | 2128 (17.3) |
| 4 | 379 485 (18.5) | 2740 (22.3) |
| 5 (most deprived) | 291 502 (14.2) | 2866 (23.3) |
| Missing | 229 558 (11.2) | 1714 (14) |
| Year of cohort entry, n (%) | | |
| 2008 | 1 060 098 (51.7%) | 6831 (55.6%) |
| 2009 | 130 610 (6.4%) | 954 (7.8%) |
| 2010 | 126 825 (6.2%) | 961 (7.8%) |
| 2011 | 160 853 (7.8%) | 1020 (8.3%) |
| 2012 | 134 390 (6.6%) | 870 (7.1%) |
| 2013 | 139 113 (6.8%) | 757 (6.2%) |
| 2014 | 119 354 (5.8%) | 503 (4.1%) |
| 2015 | 97 198 (4.7%) | 279 (2.3%) |
| 2016 | 82 962 (4.0%) | 102 (0.8%) |

Children with no recorded data on ethnicity or Townsend index were categorised as 'missing'.
IQR, IQR range.

### Crude incidence

The crude annual incidence of vitamin D prescribing in children increased by 26-fold between 2008 and 2016, rising from 10.8 (95% CI: 8.9 to 13.1) to 276.8 (95% CI: 264.3 to 289.9) per 100 000 person-years. Figure 1A shows temporal trends in vitamin D prescribing in children alongside key national guidelines or reports on vitamin

A)

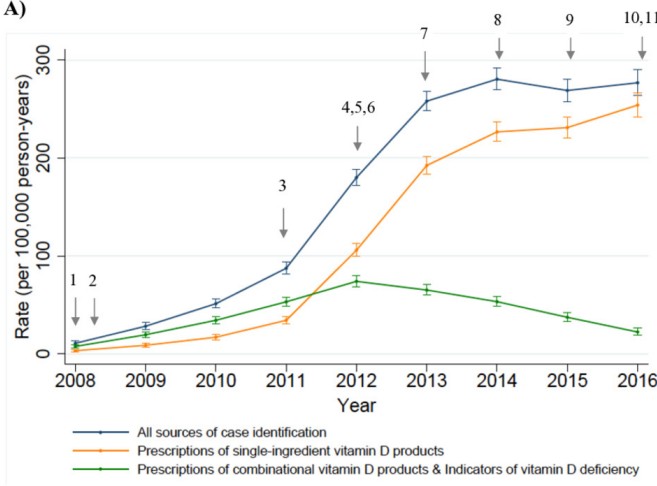

B)

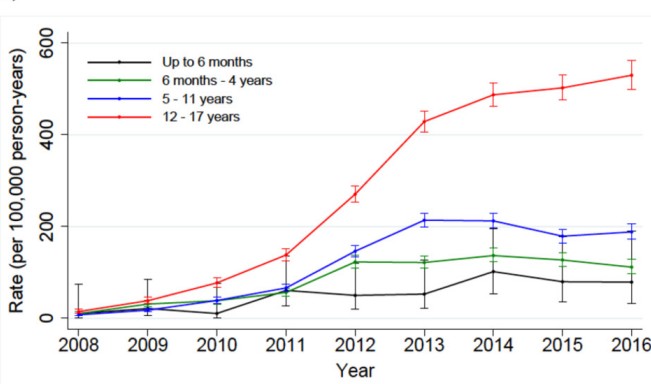

**Figure 1** (A)Time trends in vitamin D supplementation prescribing in children using all components of the case definition together and each source of case identification independently; (B) Time trends in vitamin D supplementation prescribing in children by different age groups. crude incidence rates with 95% confidence intervals represented by vertical bars. 2007: SACN: update on vitamin D[31] 2008: NICE: improving the nutrition of pregnant and breastfeeding mothers and children in low-income households[32] 2011: the National Diet and Nutrition Survey (NDNS) rolling programme: data from years 1 & 2[52] 2012: Department of Health: Vitamin D advice on supplements for at risk groups[33] 2012: British Paediatric and Adolescent Bone Group's position statement on vitamin D deficiency[34] 2012: RCPCH: vitamin D: position statement[35] 2013: RCPCH: guide for vitamin D in childhood[36] 2014: NICE: vitamin D: increasing supplement use in at-risk groups[37] 2015: NOS: vitamin D and bone health[38] 2016: SACN: vitamin D and health report[39] 2016: NICE clinical knowledge summaries: vitamin D deficiency in children[40] SACN, Scientific Advisory Committee on Nutrition; NICE,The National Institute for Health and Care Eexcellence; RCPCH, Royal College of Paediatrics and Child Health; NOS, National Osteoporosis Society

D deficiency.[31–40 52] Prescribing rates for females were higher than males, with an average ratio of male to female prescribing rate of 1:1.5 across all time periods. As can be seen from figure 1B, the increasing prescribing trend (rate per 100 000 person-years) was particularly apparent in children aged 12–17 years (14.8 (95% CI: 11.1 to 19.8) to 529.5 (95% CI: 499.2 to 561.7)), followed by those

aged 5–11 years (7.8 (95% CI: 5.4 to 11.2) to 188.3 (95% CI: 172.5 to 205.5) ], 6 months–4 years (10.2 (95% CI: 6.9 to 15.1) to 111.6 (95% CI: 96.8 to 128.7)), and less than 6 months (10.6 (95% CI: 1.5 to 75.2) to 79 (95% CI: 32.9 to 189.7)). When vitamin D prescription was defined using single-ingredient vitamin D products only, we found similar trends to the main analysis (figure 1A).

### Adjusted incidence rates

In multivariable analysis, increasing age, non-white ethnicity and social deprivation were associated with higher incidence rates of vitamin D prescribing (table 2). Among children aged 12–17 years, sex difference in incidence rate ratios was observed, with 1.8-fold higher vitamin D supplementation prescribing in females than males. General practices in England were also associated with higher IRRs as compared with Wales, Scotland and Northern Ireland. For both males and females, adjusted IRRs were higher than their corresponding crude values, with a 32-fold and 39-fold increase in the incidence of vitamin D prescribing between 2008 and 2016, respectively. A sensitivity analysis using complete cases only was comparable to the main analysis, although the magnitude of the effects of ethnicity was marginally greater for both sexes, along with a smaller temporal difference in annual incidence rates (online supplementary table S5).

### 25-hydroxyvitamin D testing and presence of symptoms at initiation of supplementation

25(OH)D concentrations recorded before starting vitamin D supplementation are presented in figure 2A. Overall, 28.7% to 56.4% of prescriptions annually had no linked 25(OH)D results recorded in the 3 months prior to the index date. The proportion of children with laboratory-confirmed vitamin D deficiency (25(OH)D<25 nmol/L) increased from 25.7% to 40.9% between 2008 and 2011 before showing a gradual decrease to 27.2% in 2016, coinciding with the issue of public health guidelines on primary prevention of vitamin D deficiency for at-risk individuals. Over the study period, increased prescribing of vitamin D supplementation was observed in children with insufficient status (25(OH)D between 25–50 nmol/L]), increasing from 15.8% to 21.7% between 2008 and 2011 to 27.9% to 37.1% during the period between 2012 to 2016.

29.5% of children had symptoms related to vitamin D deficiency (online supplementary table S5). The most commonly recorded symptoms were those related to musculoskeletal or non-specific pain, tiredness or fatigue. In the group of children with no recorded pre-supplementation 25(OH)D results, only 15.4% had recorded symptoms.

### Types of supplementation

Overall, 109 different pharmaceutical preparations of at least 32 dosage strengths (eg, 3000 IU/mL, 20 000 IU/tablet) were prescribed. A temporal change in supplementation form was observed: colecalciferol products

**Table 2** Multivariable poisson regression models of incidence of vitamin D supplementation prescribing in children stratified by sex.

| | Male | Female |
| --- | --- | --- |
| | IRR* (95% CI) | IRR* (95% CI) |
| Age group | | |
| Up to 6 months | 1.00 (Reference) | 1.00 (Reference) |
| 6 months–4 years | 1.93 (1.21 to 3.07) | 1.28 (0.84 to 1.96) |
| 5–11 years | 2.47 (1.55 to 3.92) | 2.25 (1.48 to 3.42) |
| 12–17 years | 4.15 (2.61 to 6.60) | 7.41 (4.87 to 11.27) |
| Ethnicity | | |
| White | 1.00 (Reference) | 1.00 (Reference) |
| Asian | 5.05 (4.60 to 5.54) | 4.46 (4.12 to 4.81) |
| Black | 2.83 (2.53 to 3.18) | 2.83 (2.58 to 3.10) |
| Others | 2.68 (2.20 to 3.26) | 2.72 (2.33 to 3.19) |
| Mixed | 1.93 (1.59 to 2.33) | 1.84 (1.57 to 2.16) |
| Missing | 1.48 (1.37 to 1.61) | 1.42 (1.33 to 1.52) |
| Townsend index | | |
| 1 (least deprived) | 1.00 (Reference) | 1.00 (Reference) |
| 2 | 1.03 (0.92 to 1.17) | 1.03 (0.94 to 1.14) |
| 3 | 1.18 (1.06 to 1.33) | 1.20 (1.09 to 1.32) |
| 4 | 1.37 (1.22 to 1.53) | 1.37 (1.25 to 1.51) |
| 5 (most deprived) | 1.41 (1.25 to 1.59) | 1.44 (1.31 to 1.59) |
| Missing | 1.38 (1.19 to 1.60) | 1.43 (1.26 to 1.62) |
| Country | | |
| England | 1.00 (Reference) | 1.00 (Reference) |
| Wales | 0.34 (0.25 to 0.46) | 0.33 (0.25 to 0.46) |
| Scotland | 0.24 (0.19 to 0.32) | 0.18 (0.14 to 0.23) |
| Northern Ireland | 0.24 (0.16 to 0.37) | 0.17 (0.11 to 0.27) |
| Year | | |
| 2008 | 1.00 (Reference) | 1.00 (Reference) |
| 2009 | 2.47 (1.73 to 3.51) | 2.61 (1.93 to 3.52) |
| 2010 | 3.68 (2.63 to 5.15) | 4.97 (3.75 to 6.58) |
| 2011 | 6.72 (4.88 to 9.25) | 8.04 (6.12 to 10.56) |
| 2012 | 14.60 (10.72 to 19.90) | 16.16 (12.39 to 2.08) |
| 2013 | 21.10 (15.52 to 28.67) | 22.82 (17.53 to 29.72) |
| 2014 | 25.81 (18.99 to 35.09) | 28.14 (21.60 to 36.65) |
| 2015 | 29.19 (21.44 to 39.76) | 33.53 (25.70 to 43.74) |
| 2016 | 32.27 (23.64 to 44.05) | 38.87 (29.75 to 50.80) |

*Adjusted for all variables listed in the table. The multilevel models included the general practice as a random effect.
CI, confidence interval; IRR, incidence rate ratio.

accounted for 9.9% of the prescriptions in 2010 and increased to 90.2% in 2016, explaining the trends in prescribing patterns shown in figure 1 (online supplementary figure S2).

### Doses and frequencies of administration

The EDD prescribed ranged from 27 to 160 000 IU/day (prescribed doses by age group is summarised in online supplementary table S6) with a median duration of supply

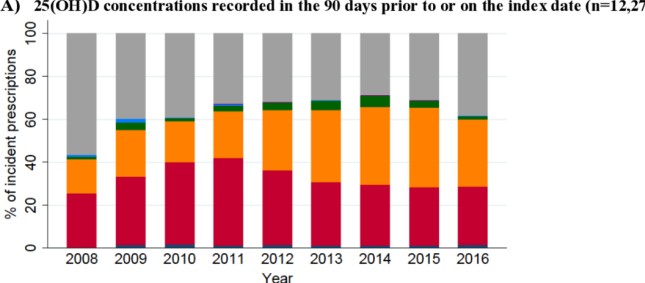

A) 25(OH)D concentrations recorded in the 90 days prior to or on the index date (n=12,277)

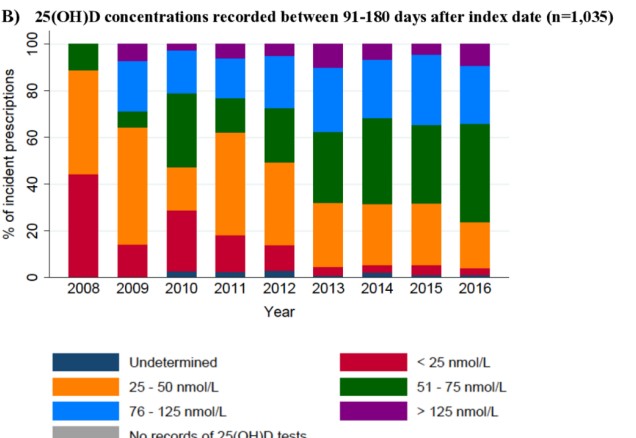

B) 25(OH)D concentrations recorded between 91-180 days after index date (n=1,035)

Legend:
- Undetermined
- < 25 nmol/L
- 25 - 50 nmol/L
- 51 - 75 nmol/L
- 76 - 125 nmol/L
- > 125 nmol/L
- No records of 25(OH)D tests

**Figure 2** (A) 25-hydroxyvitamin D concentrations recorded in the 90 days prior to or on the index date (n=12 277); (B) 25-hydroxyvitamon D concentrations recorded between 91–180 days after index date (only children with known levels, n=1035). Records with ambiguous unit of measurements were categorised as 'undetermined'.

of 56 days (IQR: 30, 83 days). There were 237 (2.0%) prescriptions with doses less than the lowest recommended primary prevention dose (<280 IU/day), and 132 (1.1%) prescriptions with doses above the highest recommended pharmacological dose (>10 000 IU/day). A total of 24 different dosing frequencies were noted, with 37.2% of children on regimens other than once-daily dosing (online supplementary table S7).

Trend analysis of supplementation regimens stratified by pre-supplementation 25(OH)D concentrations is presented in figure 3. Among children with deficiency (25(OH)D<25 nmol/L), prescriptions with EDD >1000 IU/day, considered as pharmacological doses, showed a yearly increase from 3.8% in 2008 to 79.4% in 2016 (Spearman's rho=0.983, P<0.001), which is consistent with UK recommendations. Increased used of pharmacological doses (0% in 2008 to 53% in 2016; Spearman's rho=1, P<0.001) was also noted among children with 25(OH)D concentrations between 25–50 nmol/L who, according to UK national guidance, should have been recommended a supplement at prevention doses: the corresponding temporal decrease of prescriptions at prevention doses was noted irrespective of the cut-off value used to define prevention doses (400 or ≤1000 IU). A smaller but increased trend was similarly noted in the group of children with no linked 25(OH)D concentrations (35.7% in 2008 to 49.6% in 2016; Spearman's

**A)**  No pre-supplementation 25-hydroxyvitamin D concentrations (n=3,796)

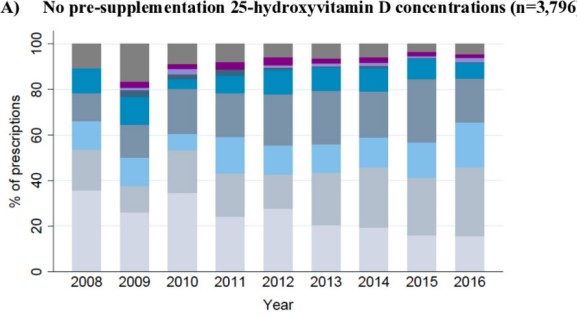

**B)**  Pre-supplementation 25-hydroxyvitamin D concentrations < 25 nmol/L (n=3,543)

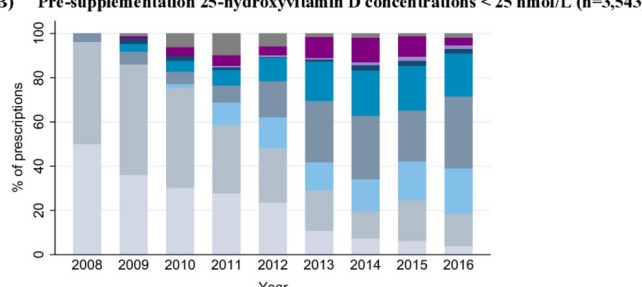

**C)**  Pre-supplementation 25-hydroxyvitamin D concentrations 25-50 nmol/L (n=3,789)

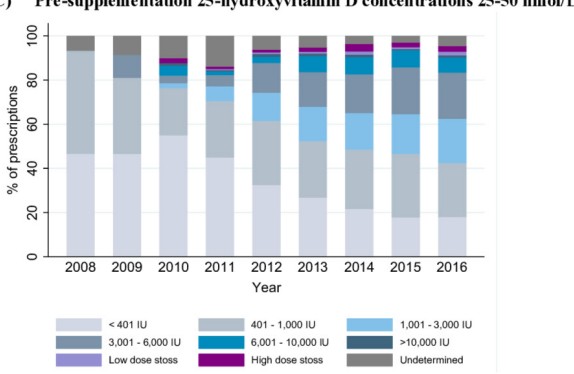

Legend:
- < 401 IU
- 401 - 1,000 IU
- 1,001 - 3,000 IU
- 3,001 - 6,000 IU
- 6,001 - 10,000 IU
- >10,000 IU
- Low dose stoss
- High dose stoss
- Undetermined

**Figure 3**  Dosages of incident vitamin D prescriptions presented in equivalent daily dose among children with: (A) no pre-supplementation 25(OH)D concentrations; (B) 25(OH)D concentrations less than 25 nmol/L; and (C) 25(OH)D concentrations between 25–50 nmol/L in the 90 days prior to their incident prescriptions.

rho=0.7333, P=0.025). Overall, vitamin D supplementation at pharmacological doses (>1000 IU/day or stoss) was prescribed to 42.8% of children with insufficient status, outside of UK recommendations.

### 25-hydroxyvitamin D testing at follow-up

In the first 90 days post-initiation of vitamin D supplementation, 9% (n=1109) of children had 25(OH)D concentrations recorded, of which 24.9% were recorded within 1 month after the index date, outside of guideline recommendations. Post-supplementation testing at the recommended time intervals was low; 25(OH)D results in the period between 91 to 180 days after incident prescriptions were available for only 8.4% (n=1035) of children (figure 2B). Over the study period, the proportion of children with post-supplementation 25(OH)D concentrations>50 nmol/L increased from 11.1% to 76.3%.

## DISCUSSION

In a large cohort representative of children in UK primary care, there has been a marked temporal increase in vitamin D prescribing, even after adjustment for changes in population demographics. The adjusted incidence rates increased by 32-fold and 39-fold between 2008 and 2016 for males and females, respectively. 25(OH)D concentrations for guiding supplementation were not available in more than 30% of children annually, and an increasing trend in prescribing vitamin D supplementation at pharmacological doses irrespective of 25(OH)D concentrations was noted with wide variations in supplementation regimens prescribed, deviating in part from UK recommendations. Furthermore, with no clear evidence to support the use of vitamin D supplementation on health outcomes, other than for the prevention or treatment of rickets and osteomalacia, this aberrant prescribing practice incurs a huge, potentially avoidable, expense to our healthcare system.

This is the first study to report UK estimates of incidence rates of vitamin D prescribing in children using a comprehensive case definition for vitamin D prescription. An earlier study based in England included only 156 general practices and reported a marked increase in the diagnosis of vitamin D deficiency and prescription costs of single-ingredient vitamin D products in children between 2000 and 2014.[28 53] Our study reports similar trends but provides data from a much larger number of practices across the whole of the UK, with detailed socio-demographic prescribing patterns and extending the data to 2016.

Vitamin D prescribing in children increased sharply between 2011 and 2014, coinciding with the publication of guidelines and reports from the UK government and national guideline committees that emphasise the importance of vitamin D supplementation in certain groups of vulnerable children and young people.[31–40] These have, in part, raised awareness of vitamin D deficiency among primary care practitioners, and explain the increase in vitamin D prescribing practices. Notably, the increase prescribing rates over time can be attributable primarily to children 5 years or older, suggesting widening awareness of vitamin D deficiency, with risk perceptions among GPs going beyond children considered as high risk. The observed change in the type of supplementation prescribed, from ergocalciferol and combination products to colecalciferol products, is congruent with increased product availability of the latter[27] and may potentially also relate to data suggesting colecalciferol is more efficacious at raising serum 25(OH)D concentrations than ergocalciferol.[54]

Patient age, sex and ethnicity were identified as drivers for prescribing, in keeping with the higher prevalence rates of vitamin D deficiency reported in teenage girls, and non-white ethnic groups in the UK.[39] Interestingly, we also observed increased rates of prescribing among practices in England compared with those in Wales (IRR=0.3), Scotland (IRR=0.2) and Northern Ireland (IRR=0.2),

suggesting other contextual factors influencing practitioners' prescribing behaviour. Disparities may be due to variations in regional strategies in implementing national guidelines, or differences in practitioners' experience and attitudes towards vitamin D deficiency, and require further exploration.[55–57]

The initiation of vitamin D supplementation should be guided by 25(OH)D concentrations, except when used for primary prevention, as indicated in UK guideline recommendations.[36 38] Yet, our results show an absence of 25(OH)D results in more than 30% of prescriptions annually since 2009, a practice also reported by others.[56] It is possible that a proportion of children may have 25(OH)D concentrations checked and recorded by hospital clinicians, although children with conditions known to affect vitamin D absorption and metabolism and thus likely to receive secondary input, were already excluded in our analysis. We also recognised that a proportion of these prescriptions may have been intended for primary prevention. However, applying the most conservative primary prevention dose definition of ≤1000 IU/day, our finding that at least 40% of prescriptions exceeded 1,000/day, and together with only a small number of children with recorded symptoms possibly related to vitamin D deficiency, would be suggestive of aberrant prescribing.

Our analysis also found an upward trend in the prescribing of vitamin D supplementation at pharmacological doses independent of 25(OH)D concentrations. To some extent, this pattern observed for children with deficiency suggests that practitioners are responsive to prescribing guidelines,[36 38] at least on a population level. On the other hand, a comparable increase was also seen in children with vitamin D insufficiency which deviates from UK national recommendations, and perhaps reflects confusion arising from multiple, sometimes conflicting, recommendations. At the same time, these deviations from recommendations might suggest a somewhat less cautious attitude towards the use of higher than recommended doses of vitamin D among practitioners. Vitamin D has a wide therapeutic window and risk of toxicity is low, nevertheless the potential risk of vitamin D toxicity exists and requires further exploration given its widespread, and sometimes unregulated use.

The major strength of our study is the large sample size that is representative of real-life clinical practice across the UK. Prescribing details in THIN are well recorded because all prescriptions from general practice are generated via a standardised computerised system. The inclusion of combination vitamin D products in the case definition also captures real-life clinical practice, but some cases could be missed if combination vitamin D product was prescribed but a diagnosis of vitamin D status was not entered in the primary care record. There are some limitations to our study too. The first vitamin D prescription recorded in THIN may not be the first ever prescription issued to the patient as initial prescribing may have occurred in secondary care and continued in general practice. However, most prescriptions issued by UK hospitals are limited to 7–14 days for clinically urgent conditions only, so the analysed sample is very likely to be representative of true practice. The study design did not allow us to determine the practitioner's rationale for prescribing, nor could we assess clinical appropriateness of prescriptions on an individual level. Trends in seasonal variations in prescribing were also not explored. Despite these limitations, our study explored the major clinical indicator for vitamin D supplementation, namely 25(OH)D concentrations and the presence of symptoms.

Our study highlights two key implications for practice. First, diverse clinical practice highlights the need for a broader understanding of the multitude of factors influencing GPs' prescribing decisions for vitamin D supplementation in order to reduce inappropriate prescribing.[55 57] Further guidelines alone, even in the absence of ambiguity, are unlikely to be effective. Second, vitamin D supplementation exceeding recommended doses, and arguably, prescriptions at prevention doses may not be the best use of the limited health resources. On this basis, at least 32.3% of our prescriptions would fall into this category. Using the most conservative estimate from an earlier study[28] (£1.65 million for the total direct cost of vitamin D prescriptions in children per year), and without costing for consultation time with healthcare professionals, dispensing or tests, we estimate an annual saving of £0.5 million.

## CONCLUSION

There has been a marked and sustained increase in vitamin D prescribing in children in UK primary care. While this may reflect the success of the Department of Health and Social Care's efforts in raising awareness of vitamin D deficiency, findings from our study would suggest that nationally set recommendations on vitamin D supplementation are not consistently followed by GPs, in terms of the number of patients treated, the doses used for supplementation, as well as the practice of prescribing vitamin D without appropriate testing. More recent NHS England guidance,[30] that recommends prescribing pharmacological doses of vitamin D supplementation only to those diagnosed with vitamin D deficiency, has the potential to reduce unnecessary prescriptions and UK healthcare expenditure, but its impact on prescribing practice would require further evaluation as previous guidelines have not always translated into practice.

**Author affiliations**
[1]Evelina Pharmacy, Guy's and Saint Thomas' NHS Foundation Trust, London, UK
[2]Institute of Pharmaceutical Science, King's College London, London, UK
[3]Research Department of Primary Care and Population Health, University College London, London, UK
[4]Population, Policy and Practice Programme, University College London, London, UK
[5]Department of Haematological Medicine, King's College Hospital Foundation NHS Trust, London, UK
[6]Renal Unit, Great Ormond Street Hospital For Children NHS Foundation Trust, London, UK

**Contributors** MW, GR, RS and JPP were involved in the conception of the study and obtaining funding. MW, LJH, EB, JPP, RS and GR contributed to the

study protocol and interpretation of data. MW had full access to the database, programming code, and performed the data extraction and analysis. MW drafted the manuscript which LJH, EB, JPP, RS and GR contributed to and revised critically before approval of the final manuscript. MW is the guarantor. The corresponding author attests that all listed authors meet authorship criteria and that no others meeting the criteria have been omitted.

**Funding** This study was funded by a Clinical Doctoral Research Fellowship grant (ICA-CDRF-2016-02-057) from the UK National Institute for Health Research (NIHR). MW is a doctoral student supported by this grant. Independent expert peer reviewers provided feedback on the grant application underpinning this study but had no further role in study design, data collection, analysis, interpretation or drafting of the manuscript. LJH is fully funded by the Wellcome Trust (209207/Z/17/Z). RS holds a Career Development Fellowship with the National Institute for Health Research. A part of the work took place in the Biomedical Research Centre at Great Ormond Street Hospital for Children NHS Foundation Trust and University College London. The views expressed are those of the authors and not necessarily those of the NHS, the NIHR or the Department of Health and Social Care.

**Competing interests** None declared.

**Patient consent for publication** Not required.

**Provenance and peer review** Not commissioned; externally peer reviewed.

**Data availability statement** Data may be obtained from a third party and are not publicly available. All data relevant to the study are included in the article or uploaded as supplementary information.

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
