## [Reviewer comments · BMJ Open]

ARTICLE DETAILS

TITLE (PROVISIONAL)	Vitamin D prescribing in children in UK primary care practices: a population-based cohort study
AUTHORS	Wan, Mandy; Horsfall, Laura; Basatemur, Emre; Patel, Jignesh; Shroff, Rukshana; Rait, Greta

VERSION 1 – REVIEW

REVIEWER	Gerald Lebovic St. Michael's Hospital, Toronto, Canada
REVIEW RETURNED	21-Jun-2019

GENERAL COMMENTS	Thank you for giving me the opportunity to review this paper. Overall, this was well written and the message came across. However, there are several issues that I think need to be addressed: 1) Methods: On page 5 lines 58-60 it is stated which children were excluded. Are there consistent codes that are used to define these conditions? This should also be updated in the STROBE table under item 6. Furthermore in this paragraph (pg. 5 - Study Population) it is unclear whether you are only using one vitamin D prescription per child (incident prescription) or you follow them longitudinally for those that have multiple prescriptions. If this is done longitudinally, this should be accounted for in the analysis using a repeated measures analysis. 2) Statistical Analysis: Page 8 lines 27-30: For the Poisson distribution was the overdispersion factor examined? If there was overdispersion how was it handled? Page 8 line 30: How did you decide what potential confounders to place in the model? Page 8 lines 39-41. Even though the interaction was significant it was removed as the quantitative impact was modest. By removing the interaction did that change the estimates of the other variables? By removing the interaction are the results and the interpretation of them very similar to the model where it was left in? Page 8 lines 54-58: A sensitivity analysis was performed using complete case data. I assume in the primary analysis there was some missing data then. How was this handled? How much data was missing? 3) Results:
---

	Last paragraph on page 11. – trend analysis (Figure 3). Is this the result from which you state in the discussion (page 14 lines 8-10) that there was an upward trend in prescribing vitamin D ...independent of vitamin D status. If so, I think this should be backed up by more than a figure. Statistically you should be able to show this upward trend for all groups. I would expect the rate of the trend is different amongst the groups (which may be interesting) but the fact that they are all increasing should be tested. Minor comments:  1) Page 6 line 45 – category is misspelled 2) Page 6 line 47, why are children with multiple ethnicity records belonging to different categories classified as missing and not as Mixed? 3) Page 7 line 49, is it worth mentioning that doses of 401-1000 are X type of dose? 4) Page 8 line 20, you mention non-normally distributed data. Was any normally distributed? If so how was this summarized? 5) Page 10 – paragraph on adjusted incidence rates. Should sensitivity analysis results be placed in a supplementary table? 6) Page 13 lines 28-36. Should you include some numbers from the comparison groups? 7) Table 1 – would be interesting to include Year here too 8) Table 2: why was age categorized and not used as a continuous variable? Comment regarding Figures: In grey scale it is difficult to visualize the different categories. Supplementary Figure S1 being an exception and was easy to understand. I suggest either using colors like those, or using some form of shading such as angled lines or symbols to distinguish the different categories.
--	---

REVIEWER	Laura Anderson McMaster University, Canada
REVIEW RETURNED	27-Jun-2019

GENERAL COMMENTS	This article describes a prospective cohort study investigating the incidence of vitamin D prescription using a large UK primary care network over a 8 year period from 2008-2016. The paper is generally well written and nicely organised and presents results that may be a valuable contribution to the literature. Major concerns:  - The context is very specific to the UK setting and needs some explanation for other readers, in particular what exactly is meant by the term “prescribing” in this context? Is vitamin D a prescription drug in the UK? It would be helpful to know more background about what is meant by “prescribed” as opposed to something just recommended by doctors. - It would be helpful if the National guidelines were better described - Could you comment on the validity of prescription data in the THIN database? - Was there any consideration of seasonal variation? Abstract  - Consider including the secondary objectives - The comparison group when stating “general practice in England” needs to be defined
---

	 - On line 40, do you mean 25(OH)D concentrations when you say vitamin D status? Or does the term “vitamin D status” refer to supplement use. It would be helpful if this was clarified throughout the manuscript. Background  - More details specific to vitamin D in children would be helpful - Include a clear description of the guidelines including introducing the definition of vitamin D deficiency earlier in the intro - First sentence in paragraph 3 could better explain what is referred to when economic burden mentioned. - The impact of vitamin D prescribing on patient benefit needs better explanation - Is it a publicly funded system? Who pays for testing? - Secondary objectives could benefit from some clarification Methods  - More information on sampling and participation in THIN would be helpful. - How valid are the measures used to define the co-morbid conditions in exclusion criteria? - A more detailed definition of “prescription” would be helpful. Would this include doctor recommendations? - More information required on the validity of prescription data in THIN would be helpful. Do you also know whether the prescriptions were filled? - Do the guidelines differ for breastfed infants in the UK? - Did you consider including season as a covariate? - Handling of missing data could be described better, including the rationale for including missing as a category in the primary analysis, and rationale for complete case analysis as sensitivity. Why not multiple imputation? - Would the term sex be more accurate than gender throughout? - Stratification not clear – more details on what was done and subsequent results needed Results  - It would be helpful to know how many children were excluded. A flow chart showing how you arrived at the final sample might be helpful. - When describing whether results aligned with recommendations it would be helpful to know which exact recommendations the authors are referring to. - Consistent language throughout the manuscript when describing dose/formulation etc. would be helpful - The tables and figures are nicely presented. Conclusions  - When considering conclusions about cost was the trade-off between less preventive prescription but more testing for deficiency considered? - Based on these results in children, it may not be appropriate to extrapolate to adults - The cost analysis presented in the discussion is not well described and may be beyond the scope of this manuscript.
--	---

VERSION 1 – AUTHOR RESPONSE

Reviewer: 1

Methods: On page 5 lines 58-60 it is stated which children were excluded. Are there consistent codes that are used to define these conditions? This should also be updated in the STROBE table under item 6.

The code lists to identify children with cystic fibrosis, chronic renal failure, inflammatory bowel diseases and chronic liver diseases were developed using published comprehensive methodology (Davé and Petersen, 2009). Code lists derived from the same method were used in an earlier publication which examined a related topic on trends in the diagnosis of vitamin D deficiency in children (Basatemur et al., 2017). The following sentence has been added to the method section: “Code lists for these conditions were created using previously published methodology”. We have also included in the manuscript that these code lists used in the study will be available on request. STROBE table has been updated accordingly.

Methods: Furthermore, in this paragraph (pg. 5 - Study Population) it is unclear whether you are only using one vitamin D prescription per child (incident prescription) or you follow them longitudinally for those that have multiple prescriptions. If this is done longitudinally, this should be accounted for in the analysis using a repeated measures analysis.

We apologise for not being clear enough on this point. In our study, we looked at incident prescriptions. To provide further clarity, we have modified the first sentence of the outcome definition accordingly as follows: “A vitamin D prescription was defined as a first prescription record (incident prescription) of either.....”.

Statistical Analysis: Page 8 lines 27-30: For the Poisson distribution was the overdispersion factor examined? If there was overdispersion how was it handled?

Thank you for this comment. We have modelled the data using both the Poisson and negative binomial regression models to examine over-dispersion. The two models were compared using Akaike’s and Schwarz’s Bayesian information criteria and prediction measures. The interpretation of both models was the same and similar estimates were provided by both models.

Statistical Analysis: Page 8 line 30: How did you decide what potential confounders to place in the model?

Thank you for this comment. Our study was intended to be exploratory and there was no pre-specified hypothesis testing. Multivariable regression analysis was used to explore the possible associations between broad demographic factors and prescribing rates, and therefore a priori decision on potential confounders was not required.

Statistical Analysis: Page 8 lines 39-41. Even though the interaction was significant it was removed as the quantitative impact was modest. By removing the interaction did that change

the estimates of the other variables? By removing the interaction are the results and the interpretation of them very similar to the model where it was left in?

Thank you for this comment. Inclusion of the interaction term did not change the estimates of the other variables (Townsend deprivation index quantile, calendar year, and country) in the model. The interpretation of both models is the same in that older age, non-white ethnicity, social deprivation and general practices in England were more likely to be prescribed vitamin D by the GP for both males and females. The incidence rates predicted by the two models were similar, but the model with the interaction term made the predictions unstable due to the same group size. We have therefore presented the model without interaction. Accordingly, we have provided further clarity in the manuscript as follows:

“A fully adjusted sex stratified model with an interaction between age and ethnicity was fitted and examined; while the interaction term was statistically significant, it did not change the estimates of the other variables and the incidence rates predicted by the two models were similar. However, as the model with the interaction term resulted in unstable predictions, the interaction term was not included in the final model.”

Statistical Analysis: Page 8 lines 54-58: A sensitivity analysis was performed using complete case data. I assume in the primary analysis there was some missing data then. How was this handled? How much data was missing?

As expected for this type of study, there were some missing data on ethnicity and Townsend deprivation index quantile. These missing data were grouped in a separate ‘missing’ category (see Table 1) and included in the primary analysis (see Table 2). We subsequently carried out a sensitivity analysis using only complete cases, and the results are provided as supplementary Table S4 in the revised manuscript as requested.

Results: Last paragraph on page 11. – trend analysis (Figure 3). Is this the result from which you state in the discussion (page 14 lines 8-10) that there was an upward trend in prescribing vitamin D ...independent of vitamin D status. If so, I think this should be backed up by more than a figure. Statistically you should be able to show this upward trend for all groups. I would expect the rate of the trend is different amongst the groups (which may be interesting) but the fact that they are all increasing should be tested.

Thank you for this comment. As our study is not hypothesis driven, the increased proportion of prescriptions at pharmacological doses over time between 2008 and 2016 was statistically confirmed using Spearman correlation coefficients.

“Among children with deficiency [25(OH)D <25 nmol/L], prescriptions with EDD >1,000 IU/day, considered as pharmacological doses, showed a yearly increased from 3.8% in 2008 to 79.4% in 2016 (spearman’s rho = 0.983, p<0.001), which is consistent in line with UK recommendations.

However, increased use of pharmacological doses (0% in 2008 to 53% in 2016; spearman's rho = 1, p<0.001) was also noted among children with 25(OH)D concentrations between 25-50 nmol/L who, according to UK national guidance, should have been recommended supplement at prevention doses; the corresponding temporal decrease of prescriptions at prevention doses was noted irrespective of the cut-off value used to define prevention doses (≤ 400 or $\leq 1,000$ IU). A smaller but increased trend was similarly noted in the group of children with no linked 25(OH)D concentrations (35.7% in 2008 to 49.6% in 2016; spearman's rho = 0.7333, p = 0.025)."

Minor comments: Page 6 line 45 – category is misspelled

Thank you for highlighting this. We have made the correction.

Minor comments: Page 6 line 47, why are children with multiple ethnicity records belonging to different categories classified as missing and not as Mixed?

We discussed the issue of ethnicity recording at length and we would ideally like to look at ethnicity in details. However, we recognised that ethnicity may be interpreted differently by different individuals at different times; it depends on the extent to which one identifies oneself with shared cultural tradition, geographical origin, ancestry, spoken language, religion and/or physical appearance. As such, it cannot be determined with certainty whether these inconsistent records are in fact a reflection of this multi-dimensional concept or if they are incorrect administrative entries. We therefore decided to take a pragmatic approach and categorised children with multiple discordant ethnicity records as missing rather than mixed for our exploratory study.

Minor comments: Page 7 line 49, is it worth mentioning that doses of 401-1000 are X type of dose?

We agree that providing information on doses 401-1000 IU in this section will provide greater clarity. Accordingly, we have included the following sentence: "Doses between 401-1000 IU were categorised as a separate prevention dose category given divergent views exist within the medical community with regards to supplementation strategies for primary prevention."

Minor comments: Page 8 line 20, you mention non-normally distributed data. Was any normally distributed? If so how was this summarized?

We only had two continuous variables which were both non-normally distributed data. As such, we have amended the sentence to "Medians and interquartile range (IQR) are presented for age at entry and follow-up time."

Minor comments: Page 10 – paragraph on adjusted incidence rates. Should sensitivity analysis results be placed in a supplementary table?

We have included the sensitivity analysis results in the supplementary section as Table S4 in the revised manuscript.

Minor comments: Page 13 lines 28-36. Should you include some numbers from the comparison groups?

We have expanded the sentence and provided numbers from the comparison groups as follows: “.....increased rates of prescribing among practices in England as compared to those in Wales (IRR = 0.3), Scotland (IRR = 0.2) and Northern Ireland (IRR = 0.2).”

Minor comments: Table 1 – would be interesting to include Year here too

Thank you for this comment. We have added year of cohort entry to Table 1 in the revised manuscript.

Minor comments: Table 2: why was age categorized and not used as a continuous variable?

The relationship between age and prescribing rate is not linear, and age was therefore specified as a categorised variable.

Figures: In grey scale it is difficult to visualize the different categories. Supplementary Figure S1 being an exception and was easy to understand. I suggest either using colors like those or using some form of shading such as angled lines or symbols to distinguish the different categories.

We apologise that the manuscript sent to you was in grey scale. We confirm that all figures were sent to the journal in colours.

Reviewer: 2

The context is very specific to the UK setting and needs some explanation for other readers, in particular what exactly is meant by the term “prescribing” in this context? Is vitamin D a prescription drug in the UK? It would be helpful to know more background about what is meant by “prescribed” as opposed to something just recommended by doctors.

- **It would be helpful if the National guidelines were better described**
- **Could you comment on the validity of prescription data in the THIN database?**

Thank you for this comment. As detailed in our response to the editorial team (see above), we have revised the manuscript to introduce UK guidance more comprehensively in the introduction section.

As to the validity of prescription data, we have provided further details on THIN data quality in the revised manuscript as follows: “Prescribing data are particularly well recorded in THIN since the computerised entry made by the GP is directly used to issue a prescription to the patient. Prescribing data in THIN has been shown to be comparable to data on dispensed prescriptions with a mean practice redemption rate for all prescribing of 98.5%.”

Was there any consideration of seasonal variation?

Thank you for this comment. Our primary objective was to examine the general temporal changes in the incidence of vitamin D supplementation prescribing over the last decade, and the inclusion of seasonal variation in the analyses was considered too detailed for this initial exploratory study. We will consider this for future studies and have added this to our study limitation.

Abstract: Consider including the secondary objectives

We understand that it is important to clearly define primary and secondary objectives. However, we have included only the main study objective in the abstract in line with the journal’s submission guidelines. In addition, including the secondary objectives would have to be at the cost of other relevant information in order to adhere to the abstract word limit.

Abstract: The comparison group when stating “general practice in England” needs to be defined

We agree that this sentence requires further context. Accordingly, we have changed the sentence to the following: “.....general practices in England (as compared to Wales, Scotland, and Northern Ireland)”

Abstract: On line 40, do you mean 25(OH)D concentrations when you say vitamin D status? Or does the term “vitamin D status” refer to supplement use. It would be helpful if this was clarified throughout the manuscript.

“Vitamin D status” in our manuscript refers to 25(OH)D concentrations. We have accordingly made changes throughout the manuscript in response to the reviewer’s comment.

Background: More details specific to vitamin D in children would be helpful

We have revised the manuscript accordingly and included the following: “In children, severe vitamin D deficiency can cause rickets and hypocalcaemic seizures, but the clinical consequences of vitamin D insufficiency are less established despite expanding literature in this field. Findings from numerous epidemiological studies have linked low 25-hydroxyvitamin D [25(OH)D] concentrations to an increased risk of a myriad of adverse health consequences in both adults and children, where low concentrations in children have been linked to asthma, eczema, respiratory tract infections, and diabetes, among others.”

Background: Include a clear description of the guidelines including introducing the definition of vitamin D deficiency earlier in the intro

As detailed in our response to the editorial team (see above), we have revised the manuscript to introduce UK guidance more comprehensively in the introduction section.

Background: First sentence in paragraph 3 could better explain what is referred to when economic burden mentioned.

As the introduction section has been revised in response to other comments, this sentence has now been removed.

Background: The impact of vitamin D prescribing on patient benefit needs better explanation

We have made changes to the introduction section to further highlight the conflicting vitamin D literature with regards to the potential benefits of vitamin D supplementation for vitamin D deficiency and insufficiency, respectively. In addition, we have included the different vitamin D supplementation approaches used for the prevention and treatment of vitamin D deficiency.

Background: Is it a publicly funded system? Who pays for testing?

In the revised manuscript, we have included information that the UK health service is publicly funded and free.

Background: Secondary objectives could benefit from some clarification

We have clarified the study objectives as follows: “We quantified temporal changes in the incidence of vitamin D prescriptions issued by GPs in a UK population-based study, and examined the proportions of prescriptions by type of supplementation, dose, dosing schedule, as well as the proportions that can be linked to 25(OH)D laboratory test results and clinical symptoms suggestive vitamin D deficiency.”

Methods: More information on sampling and participation in THIN would be helpful.

In the revised manuscript, we have included information on the following: “The THIN database contains anonymised data from 744 general practices using the Vision computer system (In Practice Systems, London, UK); data are from all patients in participating practices unless individual patients opt-out of THIN. It contains data of approximately 16 million patients which has been shown to be broadly representative of the UK population in terms of age, sex, prevalence of medical conditions, and mortality rates. As of 2015, the THIN dataset is reported to cover 6% of the UK population.”

Methods: How valid are the measures used to define the co-morbid conditions in exclusion criteria?

Please refer to our response made to reviewer 1 above.

Methods: A more detailed definition of “prescription” would be helpful. Would this include doctor recommendations?

Thank you for this comment. We agree that a more concise definition of “prescription” within the UK context would be helpful. Accordingly, we have included in the outcome section of the revised manuscript that “Consultations where GPs did not issue a prescription but recommended self-purchase of over the counter vitamin D supplement were not considered as prescriptions.”

Methods: More information required on the validity of prescription data in THIN would be helpful. Do you also know whether the prescriptions were filled?

We have responded to this comment as detailed above (please see above).

Methods: Do the guidelines differ for breastfed infants in the UK?

In the revised manuscript, we have provided more information on age-specific recommendations from national guidelines, including those for breastfed infants.

Methods: Did you consider including season as a covariate?

We have responded to this comment as detailed above (please see above).

Methods: Handling of missing data could be described better, including the rationale for including missing as a category in the primary analysis, and rationale for complete case analysis as sensitivity. Why not multiple imputation?

Thank you for this comment. We agree that multiple imputation is one approach for handling missing ethnicity data. However, it has been shown that conventional multiple imputation does not give plausible estimates of the ethnicity distribution in THIN compared to the general UK population (Pham et al., 2017; Pham et al., 2019). While new approaches to multiple imputation have been proposed, there remains a large degree of uncertainty. On this basis and given the exploratory nature of this study, we decided on a pragmatic approach to include a missing category in the primary analysis. As requested by reviewer 1, we have included the results of the complete case analysis in the supplementary section as Table S4 in the revised manuscript.

Ref:

Pham et al., Weighted multiple imputation of ethnicity data that are missing not at random in primary care databases. IJPDS. 2017; 1(1):037

Pham et al., Population-calibrated multiple imputation for a binary/categorical covariate in categorical regression models. Stat Med. 2019; 38(5): 792–808.

Methods: Would the term sex be more accurate than gender throughout?

Thank you for this comment. We have made the change as requested.

Methods: Stratification not clear – more details on what was done and subsequent results needed

We apologies for not being clear on this point. We have revised the manuscript as follows: “Cohort characteristics are presented by sex, ethnicity, socioeconomic quintile, country and calendar year as frequencies (%) for categorical data.....” and added “Proportions of prescriptions by type of supplementation, dose, and dosing schedule, linked to 25(OH)D laboratory test results and clinical symptoms suggestive vitamin D deficiency are presented as frequencies (%).”

Results: It would be helpful to know how many children were excluded. A flow chart showing how you arrived at the final sample might be helpful.

A flow chart is provided as supplementary Figure S1 in the revised manuscript.

Results: When describing whether results aligned with recommendations it would be helpful to know which exact recommendations the authors are referring to.

We have made changes to the manuscript to provide greater clarity on this issue.

Results: Consistent language throughout the manuscript when describing dose/formulation etc. would be helpful

We made a clear distinct between dose and formulation in our manuscript as the two terms refer to different prescribing elements. "Dose" refers to the amount of vitamin D to be taken at one time (e.g. 1,000 IU) whereas "formulation" relates to the different product preparations (e.g. a tablet containing 1,000 IU of vitamin D and a tablet containing 4,000 IU of vitamin D would be considered as two formulations). As some readers may not be familiar with the term "formulation", we have amended it to preparation.

Results: The tables and figures are nicely presented.

Thank you for your positive comment.

Conclusions: When considering conclusions about cost was the trade-off between less preventive prescription but more testing for deficiency considered?

Thank you for this comment. The objectives of our study did not include a full cost analysis, and it was not our intention to draw conclusions on the most cost-effective way of managing vitamin D deficiency in primary care. Our estimation on cost saving was to draw attention to the cost implication of prescribing vitamin D supplementation at pharmacological doses when lower prevention doses were recommended, as well as to use our data to generate some debates on whether general practitioners issuing vitamin D prescriptions at prevention doses was the best use of the limited health resources when prevention, as opposed to treatment, of vitamin D deficiency can arguably be managed through self-care.

Conclusions: Based on these results in children, it may not be appropriate to extrapolate to adults

Thank you for this comment. We have removed the sentence relating to extrapolation to adults in the revised manuscript.

Conclusions: The cost analysis presented in the discussion is not well described and may be beyond the scope of this manuscript.

Thank you for this comment. Cost analysis was not part of our objectives and we agree that it may be beyond the scope of this manuscript. Accordingly, we have toned down our discussions on cost.

VERSION 2 – REVIEW

REVIEWER	Gerald Lebovic St. Michael's Hospital, Canada
-----------------	--

REVIEW RETURNED	09-Sep-2019
-------------

GENERAL COMMENTS	Well written and thank you for addressing my concerns. A couple of minor points, for your consideration, regarding the statistical analysis: 1) I would mention that age was included in categories as it had a non-linear relationship. 2) With regard to modeling I want to confirm that although it was exploratory analysis, there were not multiple models (aside from the addition of the interaction and sensitivity analysis) that were tried and there were model selection techniques used that were not listed in the manuscript. Essentially, as I understand it, you explored the relationship between variables of interest and the outcome without trying many different models. 3) Minor correction - page 13 in the tracked changes draft - 5th line down in the second paragraph. Should "However, increases used..." be "However increase use..."?
---

VERSION 2 – AUTHOR RESPONSE

Reviewer: 1

I would mention that age was included in categories as it had a non-linear relationship.

Thank you for this comment. We have made the addition in the revised manuscript as follow "Age was also fitted as a categorical variable as it had a non-linear relationship."

With regard to modeling I want to confirm that although it was exploratory analysis, there were not multiple models (aside from the addition of the interaction and sensitivity analysis) that were tried and there were model selection techniques used that were not listed in the manuscript. Essentially, as I understand it, you explored the relationship between variables of interest and the outcome without trying many different models.

We confirm that we only fitted the models as detailed in the manuscript with pre-specified interactions.

Minor correction - page 13 in the tracked changes draft - 5th line down in the second paragraph. Should "However, increases used..." be "However increase use..."?

Thank you for this comment. We have amended the sentence.